



# Data on the elemental composition (mobile fractions and total content) of soils in catena at the SE Valdai Hills, Russia

Polina Enchilik, Ivan Semenkov, Nikolay Kasimov

Lomonosov Moscow State University, Faculty of Geography, Department of Landscape Geochemistry and Soil Geography

*Correspondence to: Polina Enchilik (polimail@inbox.ru)*

**Abstract.** This study presents a dataset on seasonal soils sampling from September 2016 to May 2018 in the southern part of the Central Forest Reserve (SE Valdai Hills) within a catena with Endocalcaric Albic Glossic Stagnic Profondic Retisols (Cutanic, Loamic) and Albic Gleyic Histic Retisols (Cutanic, Loamic) under coniferous-deciduous forest (*Tília cordáta, Pícea ábies, Ácer platanoídes*) on loess-like loams underlain by carbonate moraine deposits. 152 soil samples were taken to define total concentration of 67 chemical elements (ChEs), content of three mobile fractions (exchangeable, bound within organo-mineral complexes, bound with Fe and Mn hydroxides) of 69 ChEs and content of residual fraction, including macro elements (Al, Ca, Fe, K, Mg, Mn, Na, P, Ti, S, Si), heavy metals (Ba, Co, Cr, Cu, Ni, Pb, Rb, Sr, Th, U, V, Zn), trace elements (Ag, As, B, Be, Bi, Br, Cd, Cs, Ge, Hf, Li, Mo, Nb, Pd, Sb, Sc, Se, Sn, Ta, Te, Tl,W, Zr) and rare earth elements (Ce, Er, Eu, Gd, La, Lu, Nd, Pr, Sm, Tb, Tm, Dy, Ho, Y, Yb). We measured pH-value, total organic carbon content (TOC), seven particle-size classes (<1, 5-1, 10-5, 50-10, 250-50, 500-250, 1000-500 μm), and basicity from carbonates.

The dataset is available from Mendeley Data (http://dx.doi.org/10.17632/r29psg69z7.1, Enchilik et al., 2020) and will be further updated.

**Keywords.** Potentially toxic elements, taiga, landscape geochemistry, specially protected natural territories, etalon ecosystems, ecosystem monitoring

## 1 Introduction

The assessment of background state of landscapes, being basically provided in biosphere reserves, takes a special place in international programmes on the environment by UNESCO and UNEP. The analysis of geochemical structure of landscapes at different levels is an important part of background monitoring. Nowadays, the basic method is catenary. It is founded on detection of typical objects and studies of chemical compounds allocation in its components. The parameters on radial and lateral allocation of elements in different parts of landscape are assessed on the example of model catenas which include the most spread elementary landscapes and its linkings.

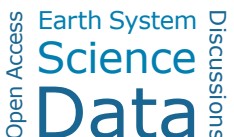

## 2 Study object

In the Central Forest State Natural Biosphere Reserve (fig.1), the most common parent rocks are loess loams underlain by carbonate Valdai glaciation moraine deposits at the depth of 90-190 cm (Karavanova & Malinina, 2007, 2009, Puzachenko et al., 2006). Studied catena is located in the southern part of the reserve on the interfluve gentle slope (<2°) with southeast exposure (fig.2, table 1) covered by, covered by coniferous-deciduous plant communities that grow in summit (1) and middle slope (2) positions and southern taiga coniferous forests that grow in the waterlogged footslope positions (3,4). GPS coordinates of soil pits are: (1) N56°27'48.7'' E32°57'45'', (2) N56°27'47.5''E32°56'15.4'', (3) N56°27'47.1'' E32°56'19.8'', (4) N56°27'48.0'' E32°56'21.1''.

Catena was chosen in the south part of reserved core alongside the transect 91/92 for monitoring of structure, dynamics and functioning of reference south taiga ecosystem monitoring systems (Puzachenko et al., 2013; Puzachenko et al., 2006).

The following changes are traced along the studied catena. Well drained summit position is a place where substances enter the ground basically from the atmosphere with wet and dry precipitation and migrate down the slope. Drainage weakens in upper footslope position and accumulation prevails. Drainage is generally affected by climatic and geological-geomorhological factors: rainfall and permeability respectively. Low permeability of parent rocks is characteristic for the territory of the reserve (Puzachenko et al., 2006) resulted in waterlogged conditions at the lower footslope position (soil profile 4) and a temporary watercourse, preferably after heavy rains. As a result, soil-moisture increases down the catena followed by the change in plant communities.

Summit (1) catena landscape is well drained that led to formation of Endocalcaric Albic Glossic Stagnic Profondic Retisols (Cutanic, Loamic) with horizons (photos of soil pits are shown in fig.3): Oi- Oe- Oa– Ah– Esc– BE– 2Bwk– 2BClk— 2Csck(l) (Barham et al., 2006; IUSS Working Group WRB, 2014) under *Tilia cordata+ Picea abes* with *Acer platanoides* and *Ulmus glabra - Corylus avellana - Oxalis acetosella* plant community *(Stellaria holostea, Anemone nemorosa, Lamium galeobdolon, Oxalis acetosella, Pteridium aquilinum, Aegopodium podagraria)*.

Albic Glossic Stagnic Profondic Retisols (Cutanic, Humic/ Ochric, Loamic) with horizons: Oi- Oe- Oa– Ah– AhE– Escl– BEscl– Bwsc– 2Bwsc(l)– 2Csck(l) under *Picea abies+Tilia cordata+Acer platanoides - Corylus avellana* – grass- plant community has developed on convex upper slope (2). *Tilia cordata* is the most frequent specie among broadleiaf, *Corylus avellana L* dominates in undergrowth. Ground cover is represented by nemoral species *(Hepatica nobilis, Galium odoratum, Lámium galeóbdolon, Ásarum europaéum, Pulmonária obscúra)*, *Pterídium aquilínum L, Equisétum sylváticum* and *Oxalis acetosélla L.*

Upper footslope (3) is occupied by *Picea abies* with *Tilia cordata* and *Acer platanoides - Vaccinium myrtillus - Sphagnum* on Albic Gleyic Histic Retisols (Cutanic, Loamic) with horizons: Hi-He-Ha-Etl–Etosc–Bwg–2Bwg–2Cgk. Grass-shrub layer is dominantly represented by *Vaccínium myrtíllus. Sphagnum sp* takes place too.

Waterlogged conditions on lower footslope (4) characterize by temporary watercourse with weakly expressed relief. There is located *Picea abies* with *Salix caprea, Tilia cordáta* and *Acer platanoides - Sphagnum with Oxalis acetosella* plant



community on Albic Gleyic Histic Profondic Retisols (Cutanic, Loamic) with horizons: Hi- He- Ha- Etoscl– BEtoscl– B– 2Bg– 2Crk.

## 3 Methods and results

### 3.1 Sampling

Samples were taken during spring (May 2018), middle summer (June 2017), middle (September 2016) and late (November 2017) autumn from the middle of each soil horizon, from the 4 pits (Fig. 2) in all seasons of the catena studied. Also, samples from the A and B horizons were collected in 9 replicates around pits no 1, 2, and 4 in June. Samples of forest litter were taken in autumn. Altogether, 152 soil samples were collected.

### 3.2 Physical and chemical properties measurement

In all samples, pH value was measured in suspension (static conditions) using a pH-meter "Expert-pH" (Russia) at the Faculty of Geography of Lomonosov Moscow State University. Total organic carbon (TOC) content was determined in 120 samples using titrimetric method with phenylanthranilic acid (Reeuwijk, 2002). The particle-size distribution in soils was analyzed using laser diffraction technique and an 'Analizeter 22' equipment (Germany). Samples for the particle-size distribution analysis were pre-treated with 4% $Na_4P_2O_7$. The Russian system of particle-size classes was used: G1 – clay (<1
μm), G2 – very fine silt (5–1), G3 – medium silt (10–5), G4 – coarse silt (50–10), G5 –fine sand (250–50), G6 – medium sand (500–250). All studied soils are loamy (containing >10% of <0,01 μm sized matter) with well-defined textural differentiation. Eluvial material is rich of silt fractions while clay fractions content is maximal in argic horizon and parent material. The content of clay fraction is also higher in parent material of middle-taiga landscapes of Karelia (Lukina et al., 2019).

### 3.3 Chemical composition measurement

Soil suspensions of NH4Ac, NH4Ac+1% EDTA, 1M HNO3 were used for mobile fraction extraction from soil subsample (soil:solution ratio of 1:5) by incubation for 18 hours. Mobile fractions (Vodyanitskii et al., 2020) F1 (exchangeable), F2 (bound within organo-mineral complexes), F3 (bound with Fe and Mn hydroxides) ChEs fraction were obtained with the use of the following reagents: F1 - with NH4Ac (ammonium acetate buffer) and the soil:solution ratio of 1:5, F2 - with 1%
EDTA (ethylenediaminetetraacetic acid) and the soil:solution ratio of 1:5 and F3 - with 1M $HNO_3$ and the soil:solution ratio of 1:10 by incubation for 18 hours. The residual fraction (F4) was determined by the difference between the total content of elements (F5) and the content of mobile forms (F1+F2+F3). Total content of chemical elements (ChEs) and mobile fractions was measured at the All-Russian Scientific-research Institute of Mineral Resources named after N.M. Fedorovsky using an

Elan-6100 ICP-MS System (Inductively Coupled Plasma Mass Spectrometer by PerkinElmer Inc., USA) and an Optima-
4300 DV ICP-AES System (Inductively Coupled Plasma Atomic Emission Spectrometer by PerkinElmer Inc., USA).

The content of elements is close to the average for Retisols of Eurasia (Semenkov et al., 2016) and agricultural soils of
Northern Europe, which were formed on moraine deposits and parent material of Scandinavia  (Reimann et al., 2018). The
content of Cr, Cu, Fe, Ni and Sr is located on the lower edge of the average for Retisols of Eurasia while the content of Mn,
Pb and Zn - on the higher edge of the average. The lowest content of Fe was found in umbric horizon of soils of summit and
upper slope landscapes due to high pH.

## 4 Description of dataset and data availability

Statistical analysis was carried out using software package 'Statistica' and Microsoft Office Excel. Descriptive statistics of
soil properties and ChEs distribution (mean, maximum, minimum, standard deviation, etc.) represented in table 2 (table S1).
This dataset is available from Mendeley Data at http://dx.doi.org/10.17632/r29psg69z7.1 (Enchilik et al., 2020). Descriptive
statistics also characterized soil properties and ChEs distribution for nine replicates of horizons A and B within each
landscape position (table S3).

The significance of seasonal changes for soil proxies was assessed using Mann-Whitney U-test (table 3, table S2). Vertical
distribution was characterized using R coefficient (Kasimov, Perelman, 1992) calculated as a ratio between the level of
elements in soil horizons to the level of elements in parent material (table S4). Spatial distribution was characterized using L
coefficient (Kasimov, Perelman, 1992) calculated as a relation between the level of elements in soil horizons of catena's
landscapes (upper slope, footslope) to the level of elements in soil horizons of the summit position (table S5). Significance of
lateral differences was assessed using Sign Test, marked tests are significant at p<0,05 (table S6). Spearman's correlation
analysis was used to calculate correlations between ChEs forms content and soil proxies. The differences  were considered
significant at p<0,05, p<0,01, p<0,001 (table S7.1, table S7.2). The calculations of relative error for elemental composition
of soil and pure extraction solutions (of NH4Ac, NH4Ac+1% EDTA, 1M HNO3) were provided as well (table S8).

## 5 Conclusion

The soils of etalon south-taiga catena were studied to define conditions of migration, content, vertical and spatial allocation
of total content and three mobile forms of its compounds: exchangeable ChEs fraction, bound within organo-mineral
complexes ChEs fraction, bound with Fe and Mn hydroxides ChEs fraction as well as mobility of these metals.
Obtaining of fundamental knowledges on differentiation of metals in the soils of south-taiga catena is necessary to assess
migration and accumulation of elements in natural and technogenic landscapes.

Reported data of the total content and mobile fractions of chemical elements in the soils of the natural landscapes might be
considered as a part of the monitoring of etalon state of south-taiga landscapes, which now is generally provided in biosphere
reserves. Data could be used by other researchers for understanding distribution of different compounds in soil catena in



soils of southern taiga forests. Data is applicable for the assessment of contamination level of the elements with potential toxicity. Data will help legislators to create health risk management plans. Elaborated data could be used for accurate identification of the sources of pollution and its migration routes as well as for more effective conservation and remediation of anthropogenically affected soils of southern taiga regions.

**Acknowledgements.** This study was supported by RSF no. 19-77-30004. The authors are grateful to L.V. Dobrydneva, A.D.
Iovcheva and E.V. Terskaya for assistance with laboratory work as well as to E.N. Aseeva, R.B. Sandlersky and Y.G. Puzachenko for assistance in field work.

**Conflict of interest.** The authors declare that they have no known competing financial interests or personal relationships that could have appeared to influence the work reported in this paper.

**Supplementary data.** The supplement related to this article is available online at Mendeley Data
(doi:10.17632/r29psg69z7.1, Enchilik, et al., 2020).

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



**Captions for Figures**

Figure 1. A - Key area location and a satellite image of Central Forest State Natural Biosphere Reserve (yellow line – protected area of the reserve and red line – core area) © Google Maps 2019.

Figure 2. Positions (hereinafter in the figures and tables): S – summit (interfluve); US – middle slope; FS – footslope. 1 – 4 Location of the soil profiles: 1 – Endocalcaric Albic Glossic Stagnic Profondic Retisols (Cutanic, Loamic), 2 – Albic Glossic Stagnic Profondic Retisols (Cutanic, Humic/ Ochric, Loamic), 3 – Albic Gleyic Histic Retisols (Cutanic, Loamic), 4 – Albic Gleyic Histic Profondic Retisols (Cutanic, Loamic). Soil horizons and materials: H – Histic, O – Folic and forest litter, A – Umbric, E – albic, B – argic, C – parent material; I – boundary between loess like loams and underlying carbonate moraine 170 deposits; II – upper boundary of effervescence with 10% HCl; III – groundwater level.

Figure 3. Photos of soil pits. Pit locations are shown in Fig. 2.

**Tables**

Table 1 Morphological properties of Retisol.

| Soil | Horizon | Depth, cm | Color | Mansell | Structure |
|---|---|---|---|---|---|
| Endocalcaric Albic Glossic Stagnic Profondic Retisols (Cutanic, Loamic) | Oa | surface | Uniform, dark brown | 10YR 4/5 | Structureless |
| | OAh | 2-6 | Uniform, dark reddish brown | 2,5YR 6/6. | Structureless |
| | E | 20-30 | Uniform, whitish grey | 10YR 6/4 | Platy |
| | | | Dark brown vertically oriented stripes of humic material | 10YR 6/6 | |
| | BE | 45-55 | Brown background | 7,5YR 6/6 | Platy |
| | | | whitish grey stripes | 10YR 7/4 | |
| | | | dark brown spots | 7,5YR 2,5/1 | |
| | 2Bwk | 70-90 | Reddish brown background | 7,5YR 6/6 | Prism-like |
| | | | clarified areas | 10YR 6/4 | |
| | 2BClk | 110-130 | Reddish brown background | 7,5YR 6/6 | Prism-like |



| | | | | | |
|---|---|---|---|---|---|
| Albic Glossic Stagnic Profondic Retisol (Cutanic, Humic/ Ochric, Loamic) | Oa | surface | Uniform, dark brown | 10YR 4/5 | Structureless |
| | OAh | 2,5-4 | Uniform, dark grey | 7,5YR 3/1 | Structureless |
| | Ah | 5-10 | Uniform, dark grey | 7,5YR 3/1 | Granular |
| | AhE | 12-18 | Light brown background whitish grey stripes | 7,5YR 4/4<br><br>7,5YR 6/1 | Platy |
| | Escl | 30-40 | Uniform, whitish grey | 10YR 7/4 | Platy |
| | BEscl | 52-62 | Brown background light brown stripes | 7,5YR 5/6<br>10YR 7/4 | Platy |
| | Bescl | 75-85 | Light whitish brown with bluish-gray stripes | 10YR 5/8<br>10YR 6/4 | Platy |
| | Bwsc | 104-114 | Reddish brown background | 5YR 5/6 | Prism-like |
| | CBwsc | 140-150 | Uniform, reddish brown | 5YR 5/6 | Granular weakly structure |
| Albic Gleyic Histic Retisols (Cutanic, Loamic) | Ha | surface | Uniform, dark brown | 5YR 2,5/1 | Structureless |
| | Ah | 9-14 | Dark reddish brown background | 2,5YR 2,5/1 | Structureless |
| | AhEtl | 17-20 | Light-gray background with light whitish brown areas | 10YR 4/1<br>10YR 5/2<br>10YR 5/3 | Prism-like platy |
| | Etosc | 25-35 | Bluish-gray background light whitish brown areas | 2,5Y 7/4<br><br>7,5YR 6/6 | Prism-like platy |
| | Bwg | 48-58 | Light brown background | 5Y 6/2<br>7,5YR 5/8 | Massive, platy |
| Albic Gleyic Histic Profondic Retisols (Cutanic, Loamic) | Ha | 0-8 | Uniform, dark brown | 10YR 2/2 | Structureless |
| | HaE | 8-12 | Light-gray background dark brown stripes | 2,5Y 6/2<br><br>2,5Y 2/1 | Massive |
| | Etoscl | 17-23 | Bluish-gray background reddish brown spots | 10YR 5/6 | Massive |



| | | | 10YR 6/3 | |
| | | | 7,5YR 3/4 | |
| Etoscl | 28-35 | Bluish-gray background reddish brown spots | 10YR 5/6 10YR 6/3 | Massive, platy |
| Betoscl | 39-45 | Bluish-gray background reddish brown spots | 10YR 6/4 | Massive, platy |
| | | | 7,5YR 7/5 | |
| Bg | 60-90 | Light brown background bluish-gray stripes | 7,5YR 5/6 | Platy |
| | | | 7,5YR 6/4 | |
| 2Crk | 110-120 | Bluish-gray tint reddish brown spots | 10YR 5/8 7,5YR 5/8 | Structureless |

Table 2. Descriptive statistics datasets

| Dataset | Purpose |
| --- | --- |
| Season/Month (all horizons, all landscape positions) | for all landscape positions and all soil horizons in certain season (May, June, September or November). |
| Landscape position (all seasons, all horizons) | for all study seasons and all soil horizons within a certain landscape position (summit, upper slope or upper and lower footslope). |
| Horizon (all seasons, all landscape positions) | for all study seasons and all landscape positions in certain soil horizon (A, El, B, C). |
| Season/Month , landscape position (all horizons) | for all soil horizons in certain season within certain landscape position. |
| Season/Month, soil horizon (all landscape positions) | for all landscape positions in certain season and soil horizon. |
| Landscape position, soil horizon (all seasons) | for all seasons within certain landscape position and in certain soil horizon. |





Earth System Science Data Discussions — Open Access

Table 3. Significance of seasonal changes

| Month | Changes | June | | | | | September | | | | | November | | | | |
|---|---|---|---|---|---|---|---|---|---|---|---|---|---|---|---|---|
| | | F1 | F2 | F3 | F4 | F5 | F1 | F2 | F3 | F4 | F5 | F1 | F2 | F3 | F4 | F5 |
| **May** | **Increases:** | $Nb_{<0,001}$ | $Na_{0,006}$ $Sr_{0,003}$ | | $Sn_{0,008}$ $Nb_{<0,001}$ | $Se_{0,004}$ | $Ti_{0,3}$ $Nb_{0,02}$ $Zr_{0,002}$ $Mo_{<0,001}$ | $Th_{0,7}$ | | $Se_{0,4}$ $V_{0,05}$ | Bi, $W_{0,003}$ | $V_{<0,001}$ | | | | $Bi_{<0,001}$ |
| | Statistically insignificant difference | Sb | As Mo U | | Mo Sb Cd | | | Mo U | As Cd | Mo | | As Sb U | Ni Pb Sr U | Cd Sb Sr | As Mo Sb | |
| | **Decreases:** | $Cr_{<0,001}$ | $Ti_{0,6}$ | | $Be_{0,04}$ Al Ca Fe K $Na_{<0,001}$ | $Ti_{0,002}$ $Be_{<0,001}$ | $W_{<0,001}$ | | $Mo_{0,02}$ Zr Ca Fe K $Na_{<0,001}$ | Al Ca Fe K Na Mg Na Se $Ti_{<0,001}$ | | | | $Be_{0,04}$ $Ti_{0,002}$ | $Al_{0,02}$ $K_{<0,001}$ | |
| **June** | **Increases:** | | | | | | Ti Nb $Zr_{<0,001}$ | $Zr_{0,02}$ $Th_{<0,001}$ | | $V_{<0,001}$ | Na $Nb_{0,002}$ | $V_{0,004}$ $Na_{<0,001}$ | | | | Ba Bi Mg $Se_{<0,001}$ |
| | Statistically insignificant difference | | | | | | | Mo | | Mo | Pb | Sb | U | | | Mo Sb |
| | **Decreases:** | | | | | | | W $Mo_{<0,001}$ $Na_{<0,001}$ | Al $Na_{<0,001}$ | $Ti_{0,004}$ K K Be | $Al_{0,001}$ K Be $Fe_{<0,001}$ K $Se_{<0,001}$ | | | $Be_{0,04}$ | $Ti_{0,01}$ | $W_{0,04}$ $Fe_{<0,001}$ |
| **September** | **Increases:** | | | | | | | | | | | $V_{<0,001}$ | | $Sn_{0,001}$ | | $Ba_{0,02}$ Mg $Se_{<0,001}$ |
| | Statistically insignificant difference | | | | | | | | | | | | | U | Cd | Mo |
| | **Decreases:** | | | | | | | | | | | | | | | $Na_{0,005}$ $K_{<0,001}$ |

subscript under a line - p-value



Figure 1

*map of Northern European Russia from Google Maps, territory zoning map from archival materials of the Central Forest Nature Reserve

Figure 2



Figure 3

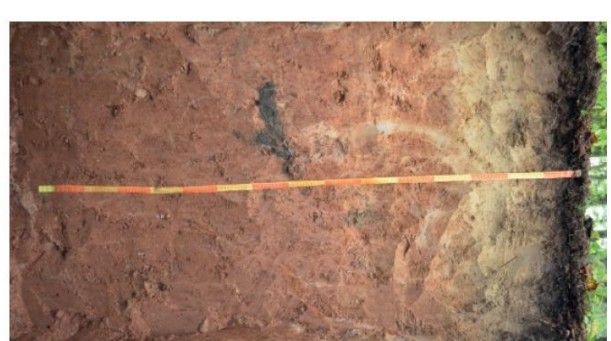

1 Summit

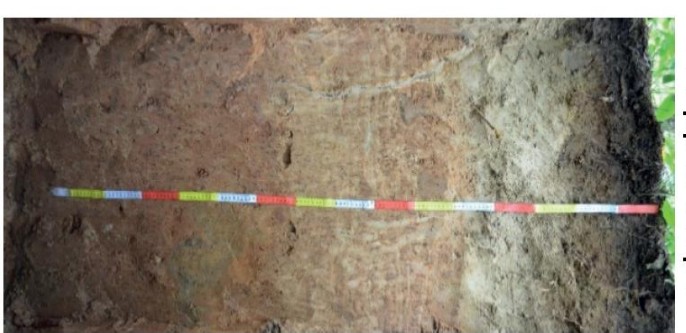

2 Upper slope

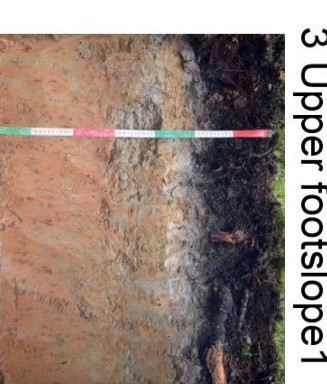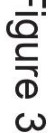

3 Upper footslope1

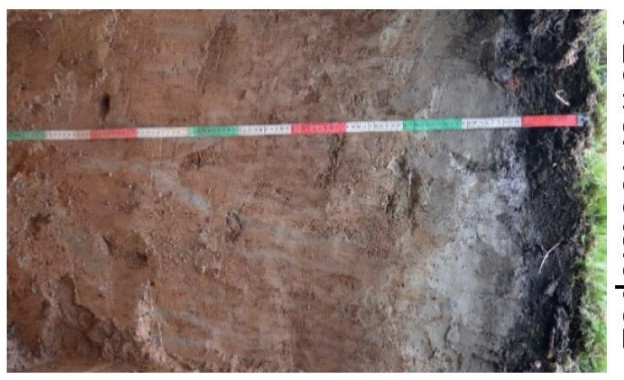

4 Lower footslope2