# Peer review of "Data on the elemental composition (mobile fractions and total content) of soils in catena at the SE Valdai Hills, Russia"

_Earth System Science Data, 2020_

## Author Comment (AC1)

Response to Reviewer # 1

We thank Reviewer #1 for thoughtful and supportive comments.

We revised the manuscript. And we hope that the revised version of the manuscript became better.

| Comment | Response |
|---|---|
| First, scientific English language is below any standards in this paper. The authors should ask a native English speaker or a professional translator to revise the text. | The revised version of the manuscript was corrected by a professional translator and by a native speaker. |
| Second, there are some disappointing mistakes in soil classification | Thank to Professor Gerasimova, the names of soils studied were corrected professionally. |
| Table 1, the first soil: Stagnic should go before Glossic, Profondic is a secondary qualifier, and thus should go after the name of the RG. | We have corrected the name of the first soil. The corrected name is Endocalcaric Albic Neocambic Stagnic Glossic Retisols (Geoabruptic, Chromic, Loamic). |
| Table 1, the second soil: Stagnic should go before Glossic, Profondica is a secondary qualifies, and the soil should be either Ochric or Himic, but not both of them. | We have corrected the name of the second soil. The corrected name is Endocalcaric Albic Neocambic Stagnic Glossic Retisols (Geoabruptic, Ochric, Lamellic). |
| Table 1, the forth soil: Profondic is a secondary qualifier. | We have corrected the name of the forth soil. The corrected name is Endocalcaric Glossic Albic Histic Stagnosols (Geoabruptic). |
| Table 1, the fifth column: please change all the commas to decimal points | We have changed all the commas to decimal points in fifth column. |
| Table 1, the sixth column: prism-like structure does not exist in any international classification, though I can imagine what You mean. Please refer to FAO or USDA manuals for field soil description | In sixth column, we have replaced prism-like structure with prismatic structure. |
| Figures 2 and 3: please try to follow internationally accepted terminology. I do not understand the expression "upper slope". Do You mean shoulder or backslope position? | We have replaced «upper slope» with «backslope» in figures 2 and 3 and throughout the text. |

| May we regard "upper footslope" and "lower footslope" as footslope and toeslope accordingly? | We have replaced «upper footslope» and "lower footslope" with «footslope» and «toeslope», respectively. |

---

## Author Comment (AC2)

Response to Reviewer #2

We thank Reviewer #2 for thoughtful and supportive comments. We hope that the revised version of the manuscript became better.

| Comment | Response |
|---|---|
| Data set problems. One of the major problems with the data set is that the exact elemental concentration values in the analyzed geochemical fractions are not provided. Only the results of some descriptive statistics, correlation analyses, statistical differences, and concentration ratios are given. Information about such statistical analyses may be informative, but they do not replace the basic data set on elemental concentration values. | We have added the exact elemental concentration values in the analyzed geochemical fractions in Table S1. |
| Additionally, the data set provided is sometimes incomplete, or it should be revised. See suggestions in the specific comments. | We have revised our manuscript in response to suggestions. See our response to the specific comments below, please. |
| The other major problem with the data set is that its uniqueness is not demonstrated in the manuscript. Based on the supported information, the reader is convinced that there is any useful potential for the data set. | The data set is unique due to a large set of soil properties (pH, the content of total organic carbon, grain-size fractions, chemical elements (including total content and the concentration of three mobile fractions), carbonates) represented for the boreal forest ecosystems studied within a toposequence. Changes in the soil properties were characterized depending on the landscape position (spatial analysis), pedon differentiation (the subsets for different soil horizons – i.e., vertical analysis), and the date of the sampling (temporal analysis). |

| | |
|---|---|
| The sampling strategy is not clear for me. The sampling was carried out on a toposequence instead of on a whole catena. | The chosen toposequence is typical for the predominant territory of the Central Forest Reserve, where spruce and coniferous-deciduous forests occupy 47% and 17% of the reserve territory, respectively (Smirnova et al., 1999). The most common parent rocks are loess-like loams underplayed by carbonate Valdai glaciation moraine deposits at a depth of 90-190 cm (Chebotareva, 1972; Puzachenko, Kozlov, 2006; Karavanova, Malinina, 2009). Drainage depends on climatic and geological-geomorphological factors. Low permeability of parent rocks is characteristic for the territory of the reserve (Puzachenko et al., 2006) resulted in waterlogged conditions at the toeslope positions and an occurrence of watercourses which appear after heavy rains. As a result, soil-moisture increases down the toposequence followed by the change in plant communities and results in the formation of downward-translocation-solutional catenas after (Sommer, Schlichting, 1997). Whole catenas with Retisols and Fluvisols of Gleysols (Urusevskaya, 1990) are rare within the reserve territory due to the flatness and waterlogging (Puzachenko et al., 2006). The studied toposequence is located in the southern part of the reserve on the interfluve gentle slope (<2º) with southeast aspect (fig.2, table 1), alongside the transect 91/92 marked to monitor the structure, dynamics and functioning of the reference south taiga ecosystems (Puzachenko et al., 2013; Puzachenko et al., 2006) that are typical for the Central Forest State Natural Biosphere Reserve and characterize drainage and distribution of substances depending on the distribution of surface water. We added this information in the revised version of the manuscript. |

| | |
|---|---|
| What suspensions were used to study soil pH? | The pH value is determined at a 1:5 soil: deionised water suspension by the potentiometric method using a pH-meter "Expert-pH" (Russia). |
| How did you analyze the CaCO3 content? | The $CaCO_3$ content is analyzed using Volumetric Calcimeter method (Standard operating procedure for soil calcium carbonate equivalent, 2020). We added this information in the manuscript. |
| The selectivity of the extractants used is questionable for the target phases. | The selectivity of the extractants used was explained in papers cited (Minkina et al., 2018; Mandzhieva et al., 2018, Anderson, 1976; Dudas, Pawluk, 1977; Whitby et al., 1978; McBratny et al., 1982; Lavado, Porcelli, 2000; Takeda et al., 2006; Torri, Lavado, 2009, Diatta, Andrzejewska, Rafałowicz, 2019). As this information was published in high-quality peer-reviewed journals available for international community, we did not repeat it in our manuscript. |
| At what pH were the extractions carried out? | The extractions was carried out with pH 4.8. |
| What organo-mineral complexes are expected to be dissolved using $NH_4Ac+EDTA$? | The difference between the metal contents in the in $NH_4Ac+1\%$ EDTA and NH4Ac extracts characterizes the content of metals weakly bound with complexes (F2) (Minkina et al., 2009, 2018). |

| | |
|---|---|
| The selectivity of 1M HNO₃ for hydrous Fe and Mn oxides must be very low. Such phases are generally extracted using a reductant and a complexing agent or a reductant together with slight acidification. But their selectivity is still very variable. The referred study (Vodyanitskii et al., 2020) also used such a method (the Tamm reagent) for hydrous Fe and Mn oxides and not 1M HNO₃. Additionally, they did not use the other two extractions to study specific operationally defined elemental fractions. | The selectivity of 1M HNO₃ for hydrous Fe and Mn oxides is acceptable (Minkina et al., 2009, 2018). Similar concentrations of HNO₃ are also used in various options for extracting mobile forms of ChEs compounds (e.g. Anderson, 1976; Dudas, Pawluk, 1977; Whitby et al., 1978; McBratny et al., 1982; Lavado, Porcelli, 2000; Takeda et al., 2006; Torri, Lavado, 2009, Diatta, Andrzejewska, Rafałowicz, 2019). |
| What digestion method was used for total element concentrations? | An open system acid digestion method was used for the dissolution of soil subsamples prior to the total elemental analysis (Karandashev et al., 2017). Along with the analyzed samples, the control samples of Gabbro Essexitovoe SGD-2A (GSO 8670-2005) were digested for the quality control. 100 mg soil subsamples were placed in 50 ml teflon beakers, 0.1 ml of a solution containing 8 mg/dm3 161Dy was added and moistened with a few drops of deionized water. Afterwards, 0.5 ml of HClO4 (Perchloric acid fuming 70% Supratur, Merck), 3 ml HF (Hydrofluoric acid 40% GR, ISO, Merck) and 0.5 ml of HNO3 (Nitric acid 65%, max. 0.0000005% Hg, GR, ISO, Merck) were added and evaporated until intense white fumes appeared. The solution was evaporated to crystal salts. Then, 2 ml of HCl (Hydrochloric acid fuming 37% GR, ISO, Merck) and 0.2 ml of 0.1 M H3BO3 solution were added and evaporated to a volume of 0.5 – 0.7 cm3. The resulting solutions were transferred into weighing bottles, with the addition of 0.1 cm3 of a solution containing 10 mg/dm3 of In (indium, used as internal standard), diluted with deionized water to 20 ml, and analyzed. 5% of all samples were measured in duplicates.We added this information in the revised manuscript. |

| | |
|---|---|
| What about the quality control and quality of the analyses? | A standard sample of Gabbro Essexitovoe SGD-2A (GSO 8670-2005) was used for quality control of soil samples. Cross-sectional samples were used to calculate the relative error. The elemental composition of blank solutions was also analyzed. |
| Did you use parallel analyses? | Yes. We used parallel extraction procedure for ChE fractionation. |
| What standard reference materials were analyzed? | We used high purity standards manufactured in Russia for preparing extraction solutions. |
| The manuscript is hard to be understood. A thorough English revision is necessary. | The revised version of the manuscript was corrected by a professional translator and by a native speaker. |
| Specific comments
Use "concentration" instead of "level" for chemical elements in the soil. | We have changed "level" on "concentration" for chemical elements in the soil. |
| L8 (and other places) I would not say seasonal sampling after four sampling campaigns on different dates. | We have changed "seasonal sampling" on "four sampling campaigns on different months". |
| L10 (and other places) "Loams" is not a petrological term. | In this case, this term is suitable. It is used in pedology by other authors e.g., Samonova, Aseyeva, 2020 (https://doi.org/10.1016/j.dib.2020.105450), Zach, Tiessen, Noellemeyer, 2006 (https://doi.org/10.2136/sssaj2005.0119) |
| L13 Do no use the term "heavy metal" for Rb, Sr, etc. Better to use trace metal and metalloids or trace elements. | We have replaced "heavy metal" with "potentially toxic elements". |
| L16 Soil "basicity" or alkalinity is measured through pH analysis. You have analyzed (?) the carbonate content of the soils. | We agree with Reviewer. We analyzed the $CaCO_3$ content of the soils using Volumetric Calcimeter method. |

| | |
|---|---|
| L23 What is the "geochemical structure of a landscape"? | Vertical and spatial flows have different geochemical features; their ratio forms the background migration geochemical structure of the landscape. We have changed this paragraph and deleted "geochemical structure of a landscape". |
| L38-39 What "substances enter the ground" at the "summit" position? Why do they not also enter lower slope positions if they are expected to enter through wet or dry deposition? | We have deleted this paragraph. |
| L39 Use "deposition" instead of "precipitation". | We have deleted this paragraph. |
| L78-79. What is the relevance of the parent material in Karelia for the study area? | We considered the Karelia as a source of material for the parent rocks of the Central Forest Reserve. We have removed this cite. |
| L98 Table 2 does not present the descriptive statistics. | Descriptive statistics of soil properties and ChEs distribution represented in table S2. Explanation of the structure of table of Descriptive statistics is represented in table 2. We made a correction. |
| L102 Element concentrations are not proxies in this case. | We have changed "proxies" on "ChEs concentrations". |
| L108 What differences do you mean? | We mean spatial differences. |
| Figure 1. A more detailed and informative location map is needed. | We have added more detailed and informative location map on figure 1. |
| Table S1. Wrong mean pH values (G9, G1425, G2133, G2841, and in many other cells). "July" appears instead of from B356 to B709 cells. | Table S2 (previous S1). We have corrected pH values and changed "July" on "June" in Table S1. |
| Table 3 and Table S2. Information is supported only for 3-10 elements in Table 3 and 14-17 elements in Table S2 for a chemical fraction. What about the other elements? The significance level is missing in several cases in Table 3. Different elements are presented in the two tables with minor overlapping. | Table S3 (previous S2). We focused on potentially toxic elements migrated in soils studied as cations (Sr, Cu, Zn, Cd, Co, Mn, Pb, Fe, Ca, Ni, Ti) and anions (As, Mo, U, Sb, Cr). |

| | |
|---|---|
| Table S3. Coarse sand, density, and concentration values in several elemental fractions (e.g., Ag1, Al1, etc.) are not given here, although these values were presented in Table S1. Why? | Table S4 (previous S3). Density and $CaCO_3$ content were not measured for table S4. We have added this information. |
| Table S8 – Relative error for what? Are they calculated from the parallel analyses? 100% for Al% fraction seems to be very high. Very bad values are provided in many other cases. What are the reference solutions? | Table S9 (previous S8) The relative errors are calculated for the total content of chemical elements and the content of elements in three extracts. They are calculated from the parallel analyses. Yes, there was an inaccuracy in the previous calculations of the relative error, now it is corrected. We have changed "reference solutions" on "blank solution". |